# Beneath the Surface: Unveiling Harmful Memes with Multimodal Reasoning Distilled from Large Language Models

**Hongzhan Lin[1], Ziyang Luo[1], Jing Ma[1]\*, Long Chen[2]**
[1]Hong Kong Baptist University
[2]The Hong Kong University of Science and Technology
{cshzlin,cszyluo,majing}@comp.hkbu.edu.hk, longchen@ust.hk

## Abstract

The age of social media is rife with memes. Understanding and detecting harmful memes pose a significant challenge due to their implicit meaning that is not explicitly conveyed through the surface text and image. However, existing harmful meme detection approaches only recognize superficial harm-indicative signals in an end-to-end classification manner but ignore in-depth cognition of the meme text and image. In this paper, we attempt to detect harmful memes based on advanced reasoning over the interplay of multimodal information in memes. Inspired by the success of Large Language Models (LLMs) on complex reasoning, we first conduct abductive reasoning with LLMs. Then we propose a novel generative framework to learn reasonable thoughts from LLMs for better multimodal fusion and lightweight fine-tuning, which consists of two training stages: 1) Distill multimodal reasoning knowledge from LLMs; and 2) Fine-tune the generative framework to infer harmfulness. Extensive experiments conducted on three meme datasets demonstrate that our proposed approach achieves superior performance than state-of-the-art methods on the harmful meme detection task.

## 1 Introduction

The development of social media platforms has given rise to a new form of multimodal content known as: **meme**. A meme typically comprises a picture that is combined with a concise text component. Memes possess the capacity to quickly spread across the internet, especially on social media platforms, due to their ease of dissemination. While memes are often seen as humorous, there is a potential for harm when the combination of images and texts is strategically used to promote political and sociocultural divisions. For instance as in Figure 1(a), during the COVID-19 pandemic, a widely

circulated meme falsely claimed that the mRNA vaccine would alter human genetic code (DNA)[1]. Such multimodal disinformation spread caused vaccine safety and effectiveness concerns, hindering the formation of strong immune defenses in impacted areas globally (Basch et al., 2021; Lin et al., 2022). Besides, another meme example shown in Figure 1(b) perpetuates harmful stereotypes and generalizations about Asians. Therefore, it is necessary to develop automatic approaches to facilitate harmful meme detection for unveiling the dark side of memes.

Harmful memes[2] are generally defined as "multimodal units consisting of an image and accompanying text that has the potential to cause harm to an individual, an organization, a community, or the whole society" (Sharma et al., 2022). Previous studies (Kiela et al., 2020; Pramanick et al., 2021a,b) attempted to straightforwardly utilize pretrained vision-language models (Li et al., 2019; Lu et al., 2019) for harmful meme detection by training additional task-specific classification layers. More recently, Cao et al. (2022) proposed a prompt-tuning method with the meme text and image caption as the prompt for masked language modeling (Devlin et al., 2019; Liu et al., 2019).

However, existing harmful meme detection approaches oversimplified the problem as an end-to-end classification paradigm, which only recognizes the superficial signals conveyed through the surface text and image. But more in-depth investigation and cognition on the implicit meaning is required especially when the image and text are not obviously correlated (Pramanick et al., 2021b). Intuitively, the key to harmful meme detection is to excavate rich correlations beneath the surface

---

\* Jing Ma is the corresponding author. The first two authors contributed equally to this work.

[1]https://www.bbc.com/news/55101238

[2]**Disclaimer**: *This paper contains discriminatory content that may be disturbing to some readers, where meme examples and words are offensive or hateful in nature. These contents are provided for illustrative purposes only and do not represent the views and standpoints of the authors.*

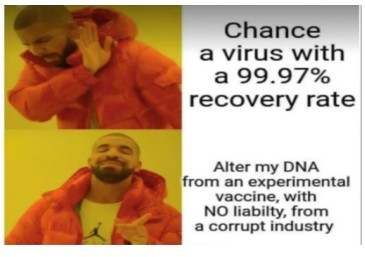 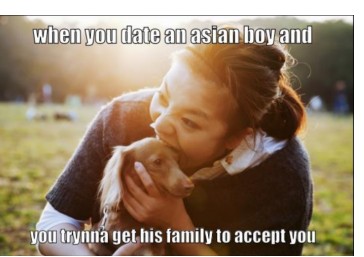 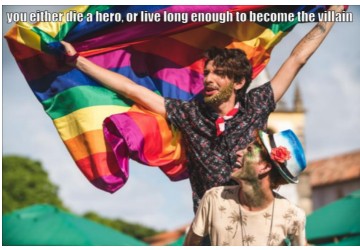

|(a) Harmful|(b) Harmful|(c) Harmless|

Figure 1: Examples of harmful and harmless memes. **Meme text**: (**a**) *Chance a virus with a 99.97% recovery rate; Alter my DNA from an experimental vaccine, with NO liability, from a corrupt industry.* (**b**) *when you date an asian boy and you trynna get his family to accept you.* (**c**) *you either die a hero, or live long enough to become the villain.*

of the seemly uncorrelated text and image in the meme: 1) For example as in Figure 1(b), the image and the text are not harmful when considered in isolation, but are harmful when taken as a whole. A human checker should cognize that, the "biting" action in the image of a young woman with her pet dog ridicules Asians' "dog-eating" behavior, which corresponds to the "asian" word in the text. 2) In contrast, some harmful signals (*e.g.*, "die" or "villain") are observed in the text of Figure 1(c), but the meme itself actually does not promote hate or discrimination against a particular group of people. Because the text is a quote from a popular movie and is often used as a philosophical statement about the choices people make in life. And the image further adds a celebratory and joyful tone to the overall message. In comparison, conventional detection methods just focused on recognizing shallow harm-indicative signals without such multimodal reasoning and essential background knowledge consideration, so the social dynamics of different races or the origin of the meme text from classical movie lines may not be well-cognized. Unlike such recognition-level detection models, we argue that establishing reasonable thought between textual and visual information can further improve meme understanding with background knowledge for better harmful meme detection.

Inspired by the success of LLMs for reasoning at the cognition level with contextual background knowledge (Wei et al., 2022; Kojima et al., 2022; Zhang et al., 2022), we propose a novel approach: **MR.HARM**, by leveraging the **M**ultimodal **r**easoning knowledge distilled from LLMs for **Harm**ful meme detection. To this end, we first prompt LLMs for abductive reasoning, and then propose a two-stage generative framework based on smaller language models to learn reasonable thoughts from LLMs for better multimodal fusion

and lightweight fine-tuning. More specifically, we incorporate the meme text and image into a **two-stage** training paradigm: 1) *Reasoning Distillation*: In the first stage, we fine-tune our smaller language models with the interaction of language and vision features to distill multimodal reasoning knowledge from LLMs, which empowers our framework with the ability to conduct cognitive reasoning for the harmfulness prediction. 2) *Harmfulness Inference*: In the second stage, we exploit the fine-tuned small language models to infer the final harmfulness prediction. In this manner, we augment the harmful meme detection model with multimodal reasoning knowledge to unmask the implicit meaning hidden in holistic multimodal information from memes.

We evaluate our proposed approach based on three public meme datasets. The results not only show that our method outperforms strong harmful meme detection baselines by a large margin, but also provide fine-grained analysis for interpreting how our approach works. Our contributions are summarized as follows in three folds:

- To our best knowledge, we are the first to alleviate the issue of superficial understanding for harmful meme detection by explicitly utilizing commonsense knowledge, from a fresh perspective on harnessing advanced LLMs.[3]

- We propose a novel generative framework to fine-tune smaller language models augmented with the multimodal reasoning knowledge distilled from LLMs, which facilitates better multimodal fusion and lightweight fine-tuning for harmfulness prediction.

- Extensive ablations on three meme datasets confirm that our method could yield superior

---

[3]Our code is available at https://github.com/HKBUNLP/Mr.Harm-EMNLP2023

performance than state-of-the-art baselines for the harmful meme detection task.

## 2 Related Work

### 2.1 Harmful Meme Detection

Harmful meme detection is a rapidly growing area in the research community, driven by the recent availability of large meme benchmarks (Kiela et al., 2019; Suryawanshi et al., 2020; Pramanick et al., 2021a). The Hateful Memes Challenge organized by Facebook (Kiela et al., 2020) further encouraged researchers to develop solutions for detecting harmful memes in hate speech (Das et al., 2020). More recently, Pramanick et al. (2021a) firstly defined the harmful meme concept and demonstrated its dependence on contextual factors. The complex nature of memes, which often rely on multiple modalities, makes them challenging and struggle to yield good performance only using unimodal detection methods (Simonyan and Zisserman, 2014; He et al., 2016; Devlin et al., 2019). Therefore, recent studies in this area attempted to apply multimodal approaches on the harmful meme detection task.

Previous studies have employed classical two-stream models that integrate text and vision features, which are learned from text and image encoders, using attention-based mechanisms and multimodal fusion techniques for classifying harmful memes (Kiela et al., 2019, 2020; Suryawanshi et al., 2020). Another branch was to fine-tune pre-trained multimodal models specifically for the task (Lippe et al., 2020; Muennighoff, 2020; Velioglu and Rose, 2020; Hee et al., 2022). Recent related efforts have also sought to explore the use of data augmentation techniques (Zhou et al., 2021; Zhu et al., 2022), ensemble methods (Zhu, 2020; Velioglu and Rose, 2020; Sandulescu, 2020) and harmful target disentanglement (Lee et al., 2021). More recently, Pramanick et al. (2021b) proposed a multimodal framework by using global and local perspectives to detect harmful memes which achieves state-of-the-art performances. A follow-up prompt-based approach (Cao et al., 2022) attempted to concatenate the meme text and extracted image captions to fine-tune masked language models (Liu et al., 2019) for harmful meme detection. However, existing solutions only capture the superficial signals of different modalities in memes in an end-to-end manner, which largely ignore explicit deductive reasoning to guide the model for understanding background knowledge about the complex and diverse relations between the visual and textual elements.

### 2.2 Large Language Models

Recently, LLMs have demonstrated remarkable capability in complex reasoning (Brown et al., 2020; Thoppilan et al., 2022; Rae et al., 2021; Chowdhery et al., 2022), such as generating intermediate inference procedures before the final output (Nye et al., 2021; Wei et al., 2022; Kojima et al., 2022; Zhang et al., 2022). Unfortunately, the large size of LLMs restricts their deployment on detecting harmful memes with different modalities, regardless of how they are enhanced with strategetic text prompting. Knowledge distillation has been successfully used to transfer knowledge from larger, more competent teacher models into smaller student models affordable for practical applications (Buciluǎ et al., 2006; Hinton et al., 2015; Beyer et al., 2022). However, existing researches on knowledge distillation from LLMs (Wang et al., 2022; Ho et al., 2022; Magister et al., 2022) only consider the language modality, they are not suitable for harmful meme detection because harmful memes can convey holistic synergy information through multimodal features. In this work, we conduct abductive reasoning with LLMs, which further advocates a multimodal reasoning paradigm to fine-tune smaller language models (LMs) for harmful meme detection.

## 3 Our Approach

**Problem Statement**   We define a harmful meme detection dataset as a set of memes where each meme $M = \{y, \mathcal{I}, \mathcal{T}\}$ is a triplet representing an image $\mathcal{I}$ that is associated with a text $\mathcal{T}$, and a ground-truth harmfulness label $y \in \{\texttt{harmful}, \texttt{harmless}\}$. In this work, to investigate multimodal reasoning distilled from LLMs, we convert the harmful meme detection task into a natural language generation paradigm, where our model takes the text $\mathcal{T}$ and image $\mathcal{I}$ as the input and generates a text sequence that contains the label $y$ to clearly express whether the meme is harmful.

Our core idea is to reason and evolve with the cognition-level rationale beyond the recognition-level perception (Davis and Marcus, 2015) by capturing the inter-relationship between visual and textual elements in memes. The overview of our framework is shown in Figure 2, which consists of abductive reasoning with LLMs (see Sec. 3.1) and two training stages, *i.e.*, reasoning distillation (see Sec. 3.2) and harmfulness inference (see Sec. 3.3).

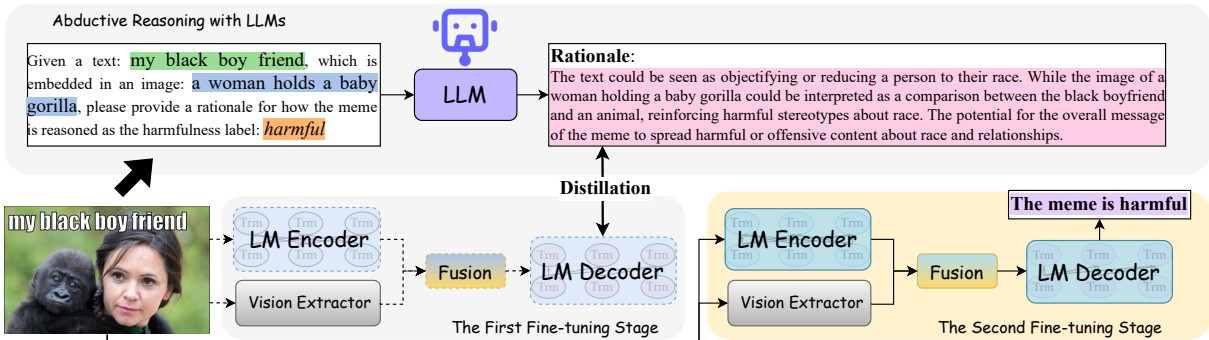

Figure 2: The overall pipeline of our method. We first conduct abductive reasoning with LLMs to extract harmfulness rationales (pink) by the prompt consisting of the meme text (green), the image caption (blue), and the label (orange). We then use the generated rationales to train small task-specific models with multimodal inputs as the first fine-tuning stage and feed the same inputs to the updated model for harmfulness inference as the second fine-tuning stage.

## 3.1 Abductive Reasoning with LLMs

In this paper, we propose to utilize abductive reasoning with multimodal inputs to train smaller downstream models. LLMs can produce natural language rationales unveiling the implicit meaning beneath the surface of the memes to justify the reason why the meme is harmful or not. This shares a similar intuition as heuristic teaching (Pintrich and Schunk, 2002) where a teacher who has rich experience and knowledge can impart to students the correct way of thinking and reasoning based on questions with corresponding answers. The students then learn how to deduce their own ways to the correct answers from questions accordingly. Thus we aim to activate explicit reasoning knowledge in LLMs as a teacher model, *e.g.*, contextual and cultural information related to memes, to guide our model to strengthen harmfulness prediction.

Given a meme sample $M = \{y, \mathcal{I}, \mathcal{T}\}$ from the training data, to prompt large language models in uniform language modality, we first extract the text caption $\tilde{\mathcal{I}}$ of the image $\mathcal{I}$ by off-the-shelf captioning models (Mokady et al., 2021). Then we curate a template $p$ that consists of a triplet $\{y, \tilde{\mathcal{I}}, \mathcal{T}\}$ as observed attributes, to prompt the LLMs to generate a rationale $r$ that elicits the reasoning knowledge about how to infer the harmfulness label $y$ based on the interplay of the meme text $\mathcal{T}$ and the image caption $\tilde{\mathcal{I}}$ as illustrated in Figure 2. Specifically, we design $p$ as:

*"Given a Text: [$\mathcal{T}$], which is embedded in an Image: [$\tilde{\mathcal{I}}$]; and a harmfulness label [$y$], please give me a streamlined rationale associated with the meme, without explicitly indicating the label, for how it is reasoned as [$y$]."*

As we clarify the ground-truth harmfulness label in the observed attributes of the prompt, the hallucination issue (Bang et al., 2023) of LLMs could be effectively alleviated. Because the rich contextual background knowledge could be activated by abductive reasoning based on the ground truth and invalid rationales are naturally filtered out.

## 3.2 Reasoning Distillation with Small LMs

Since we utilize image captions to represent the meme images, we could perform abductive reasoning with large language models pre-trained with language modality. However, only using the captions as opposed to original vision features may suffer from a lack of mutual synergy in the representation space of different modalities in memes due to the inductive bias of possible information loss in the captioning process. On the other hand, LLMs can be used to conduct abductive reasoning only for the training data whose harmfulness label is given in prior but is challenging to be fine-tuned for this task due to the huge amount of model parameters. To facilitate the interactions between the meme text and the image, we propose to fine-tune a smaller language model for the harmful meme detection task, which allows flexibility in adjusting model architectures to incorporate multimodal features and is more lightweight for task-specific fine-tuning.

In this section, we train a small language model as a student model distilled from the LLMs with multimodal reasoning knowledge. Specifically, we leverage generated rationales from LLMs as informative supervision, to fine-tune a smaller pre-trained language model to excavate the rich inter-relationship between language and vision modali-

ties of memes.

**Encoding** For a meme sample $M$ from the training data, we first encode the text $\mathcal{T}$ and the image $\mathcal{I}$ to obtain their embedding vectors as follows:

$$
\begin{aligned}
H_\mathcal{T}^0 &= \text{TE}(\mathcal{T}), \\
H_\mathcal{I} &= \text{VE}(\mathcal{I}),
\end{aligned}
\tag{1}
$$

where $\text{TE}(\cdot)$ denotes the text embedding layer of the LM Encoder. And $H_\mathcal{T}^0 \in \mathbb{R}^{m \times d}$ is the token embeddings in the Transformer encoder (Vaswani et al., 2017) where $m$ is the text token length and $d$ is the dimension of the hidden states. $\text{VE}(\cdot)$ is the Vision Extractor implemented as frozen pre-trained vision Transformers (Radford et al., 2021) to fetch the patch-level features of the image with $n$ patches, which is projected into the visual representations $H_\mathcal{I} \in \mathbb{R}^{n \times d}$. Next, to support semantic alignment between the text and the image for better context understanding, we exploit a cross-attention mechanism (Luo et al., 2022) for multimodal fusion of the textual and visual information:

$$
\begin{aligned}
Q_\mathcal{T} &= W_Q^i H_\mathcal{T}^i + b_Q^i, \\
K_\mathcal{I} &= W_K^i H_\mathcal{I} + b_K^i, \\
V_\mathcal{I} &= W_V^i H_\mathcal{I} + b_V^i, \\
H_\mathcal{I}^i &= \text{softmax}\left(\frac{Q_\mathcal{T} K_\mathcal{I}^\top}{\sqrt{d_k}}\right) V_\mathcal{I},
\end{aligned}
\tag{2}
$$

where $H_\mathcal{T}^i$ is the input hidden states of each LM Encoder layer and $H_\mathcal{I}^i$ is the attended visual features. Then we can fuse $H_\mathcal{I}^i$ with $H_\mathcal{T}^i$ to attain the interplay representations for a meme:

$$
H_\mathcal{T}^{i+1} = \text{LME}^i\left(H_\mathcal{T}^i\right) + W_O^i H_\mathcal{I}^i + b_O^i,
\tag{3}
$$

where $\text{LME}^i(\cdot)$ is the $i$-th layer of the LM Encoder, $W_*^i$ denotes the linear projection, $b_*^i$ is the bias, and $\hat{H} = H_\mathcal{T}^L$ is the final interplay representations after going through an L-layer LM Encoder fused with the visual features.

**Decoding** Finally, we feed the interplay representations $\hat{H} \in \mathbb{R}^{m \times d}$ into the LM Decoder, implemented as a Transformer-based decoder, to generate the reasonable rationale. Overall, the smaller language model $f$ is trained by minimizing the following distillation loss:

$$
\mathcal{L}_{distill} = \text{CE}\left(f(\mathcal{I}, \mathcal{T}), r\right),
\tag{4}
$$

where $\text{CE}(\cdot)$ denotes the cross-entropy loss (Sutskever et al., 2014) between the predicted text and the target rationale $r$ generated by LLMs. In this way, multimodal reasoning knowledge about the meme could be explicitly distilled from

LLMs and injected into the smaller language model specific to harmful meme detection.

### 3.3 Harmfulness Inference

During the first fine-tuning stage, we conducted explicit deductive reasoning to empower our model with the capability of multimodal reasoning distilled from LLMs. As the goal of this task is to determine whether the meme is harmful or not, we conduct the second fine-tuning stage for Harmfulness Inference, which shares the same model architecture, parameters, and encoding procedure as Sec. 3.2 but differs in the decoding output. To make the output consistent with harmfulness prediction, the smaller model $f$ is further trained by minimizing the following inference loss:

$$
\mathcal{L}_{infer} = \text{CE}\left(f(\mathcal{I}, \mathcal{T}), y\right),
\tag{5}
$$

where the cross-entropy loss is computed between the generated text and ground-truth harmfulness label $y$. With the generative objective (Raffel et al., 2020) adapted to the previous Reasoning Distillation stage, the prior reasoning knowledge absorbed in Reasoning Distillation could be well induced for Harmfulness Inference.

**Model Training** The model training consists of two fine-tuning stages: 1) Reasoning Distillation and 2) Harmfulness Inference, where Reasoning Distillation is the predecessor fine-tuning phase of Harmfulness Inference. Note that for model testing, we directly input the test sample into our fine-tuned language model to predict the meme harmfulness.

## 4 Experiments

### 4.1 Experimental Setup

**Datasets** We use three publicly available meme datasets for evaluation: (1) Harm-C (Pramanick et al., 2021a), (2) Harm-P (Pramanick et al., 2021b), and (3) FHM (Kiela et al., 2020). Harm-C and Harm-P consist of memes related to COVID-19 and US politics, respectively. FHM was released by Facebook as part of a challenge to crowd-source multimodal harmful meme detection in hate speech solutions. Different from FHM that each meme was labeled as *harmful* or *harmless*, Harm-C and Harm-P were originally labeled with three classes: *very harmful*, *partially harmful*, and *harmless*. For a fair comparison, we merge the *very harmful* and *partially harmful* memes into *harmful* ones, following the evaluation setting of recent work (Pramanick et al., 2021b; Cao et al., 2022). We provide

| Dataset | Harm-C | | Harm-P | | FHM | |
|---|---|---|---|---|---|---|
| Model | Accuracy | Macro-$F_1$ | Accuracy | Macro-$F_1$ | Accuracy | Macro-$F_1$ |
| Text BERT (Devlin et al., 2019) | 70.17 | 66.25 | 80.12 | 78.35 | 57.12 | 41.52 |
| Image-Region (He et al., 2016) | 68.74 | 62.97 | 73.14 | 72.77 | 52.34 | 34.19 |
| Late Fusion (Pramanick et al., 2021a) | 73.24 | 70.25 | 78.26 | 78.50 | 59.14 | 44.81 |
| MMBT (Kiela et al., 2019) | 73.48 | 67.12 | 82.54 | 80.23 | 65.06 | 61.93 |
| VisualBERT COCO (Li et al., 2019) | 81.36 | 80.13 | 86.80 | 86.07 | 61.48 | 47.26 |
| ViLBERT CC (Lu et al., 2019) | 78.70 | 78.09 | 87.25 | 86.03 | 64.70 | 55.78 |
| MOMENTA (Pramanick et al., 2021b) | 83.82 | 82.80 | **89.84** | 88.26 | 61.34 | 57.45 |
| MaskPrompt (Cao et al., 2022) | 84.47 | 81.51 | 88.17 | 87.09 | 72.98 | 65.24 |
| MR.HARM | **86.16** | **85.43** | 89.58 | **89.57** | **75.40** | **75.10** |

Table 1: Harmful meme detection results on three datasets. The accuracy and macro-averaged F1 score (%) are reported as the metrics. The best and second results are in bold and underlined.

statistics of the three datasets in the Appendix.

**Baselines** We compare MR.HARM with several state-of-the-art harmful meme detection systems: 1) **Text BERT** (Devlin et al., 2019); 2) **Image-Region** (Ren et al., 2016; He et al., 2016); 3) **Late Fusion** (Pramanick et al., 2021a); 4) **MMBT** (Kiela et al., 2019); 5) **VisualBERT COCO** (Li et al., 2019; Lin et al., 2014); 6) **ViL-BERT CC** (Lu et al., 2019; Sharma et al., 2018); 7) **MOMENTA** (Pramanick et al., 2021b); 8) **MaskPrompt** (Cao et al., 2022). We use the accuracy and macro-averaged F1 score as the evaluation metrics. More implementation details and baseline descriptions are provided in Appendix.

### 4.2 Harmful Meme Detection Performance

Table 1 shows the performance of our proposed method versus all the compared methods on the Harm-C, Harm-P and FHM datasets. It is observed that 1) The performance of the baselines in the first group is obviously poor due to only unimodal features like text-only or image-only being captured. In comparison, the other baselines exploit the multimodal features from both the text and image in memes. 2) The multimodal models in the second group outperform the unimodal ones. The early-fusion models with multimodal pre-training (*i.e.*, VisualBERT COCO and ViLBERT CC) outperform that of the simple fusion with unimodal pre-training (*i.e.*, Late Fusion and MMBT) on Harm-C/P datasets, while MOMENTA performs best in the second group by considering global and local information of memes. 3) However, as the images in FHM dataset are more informative and high-quality, MaskPrompt yields the best performance among all the baselines by incorporating additional

extracted entities and demographic information of the image into the masked language models, besides just captioning the image into the prompt.

Our proposed MR.HARM improves over the best baselines by 2.63%, 1.31%, and 9.86% in terms of Macro-F1 score on Harm-C, Harm-P, and FHM datasets, respectively. We observe that 1) the improvement on the Harm-P dataset is relatively milder than that on the other two datasets. Meanwhile, all the baselines just have tiny differences among their performances on Harm-P. We speculate the reason falls into the smaller dataset scale of Harm-P which only contains politics-related harmful memes. 2) A similar trend can also be observed in Harm-C and FHM datasets: the more challenging the dataset is, the greater performance improvement MR.HARM achieves. Our model performs flexibly and stably across all datasets with its keen judgment on harmful memes. This is because all the baselines are only designed at the recognition level, but MR.HARM is further empowered with multimodal reasoning knowledge distilled from LLMs to unearth harmful content from the seemly uncorrelated text and image modalities of memes.

### 4.3 Ablative Study

We perform ablative studies on several variants of MR.HARM: 1) *w/o Reasoning Distillation*: Simply fine-tune the smaller language models in the stage of Harmfulness Inference without the stage of Reasoning Distillation based on LLMs; 2) *w/o Visual Features*: Discard the features from the meme image while keeping those from the meme text; 3) *w/o Multimodal Fusion*: Instead of the fusion mechanism on the multimodal features in our language model, we only append the lingual features from image captioning together with the meme text

| Dataset | Harm-C | | Harm-P | | FHM | |
|---|---|---|---|---|---|---|
| Model | Accuracy | Macro-$F_1$ | Accuracy | Macro-$F_1$ | Accuracy | Macro-$F_1$ |
| MR.HARM | 86.16 | 85.43 | 89.58 | 89.57 | 75.40 | 75.10 |
| w/o Reasoning Distillation | 83.33 | 81.44 | 88.17 | 88.17 | 73.60 | 73.41 |
| w/o Visual Features | 82.48 | 80.30 | 87.04 | 87.03 | 58.80 | 57.01 |
| w/o Multimodal Fusion | 79.38 | 75.36 | 87.46 | 87.45 | 74.40 | 74.25 |
| w/o Two-stage Training | 83.05 | 81.45 | 63.32 | 63.32 | 67.40 | 65.77 |
| w/o Fine-tuning Small LMs | 71.75 | 66.86 | 61.13 | 60.27 | 60.00 | 57.72 |

Table 2: Ablation studies on our proposed framework.

during encoding; 4) *w/o Two-stage Training*: Concatenate the rationales generated from LLMs and golden harmfulness label as the target for model training, to replace the two-stage training paradigm; 5) *w/o Fine-tuning Small LMs*: Directly prompt the representative large language model ChatGPT based on InstructGPT (Ouyang et al., 2022) for harmful meme detection.

As demonstrated in Table 2, the ablative models suffer different degrees of performance degradation, indicating the effectiveness of our proposed components for harmful meme detection with multimodal reasoning distilled from LLMs. Specifically, the performance of MR.HARM significantly decreases in the '*w/o Reasoning Distillation*' setting due to the lack of multimodal reasoning knowledge transferred from LLMs about the seemly uncorrelated modalities in memes. The '*w/o Visual Features*' setting also achieves worse performance than MR.HARM, suggesting that the visual representations are complementary to the meme text for harm-indicative pattern extraction in the language model. MR.HARM makes improvements over '*w/o Multimodal Fusion*', which implies the promoting role of our fusion mechanism that incorporates original vision features into the language model, hardly compromised when there could be severe information loss in the captioning process. Moreover, the '*w/o Two-stage Training*' setting leads to large-margin performance degradation, which verifies the effectiveness of our two-stage training paradigm. This is because this setting causes mutual interference between intermediate reasoning and final prediction, which affects the convergence effect of harmfulness inference and damages the model's performance and stability. Compared with MR.HARM, the performance of '*w/o Fine-tuning Small LMs*' also significantly decreases, highlighting the importance of abductive reasoning with LLMs to alleviate the hallucination issue during

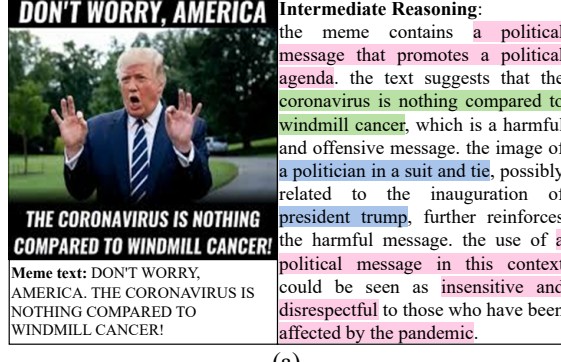

(a)

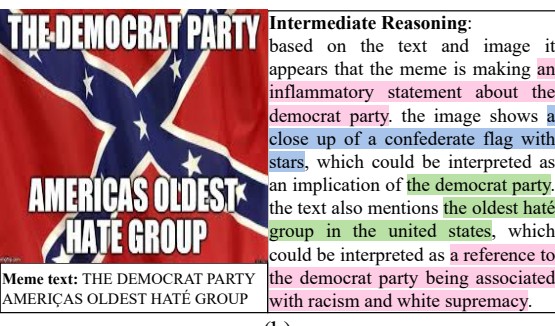

(b)

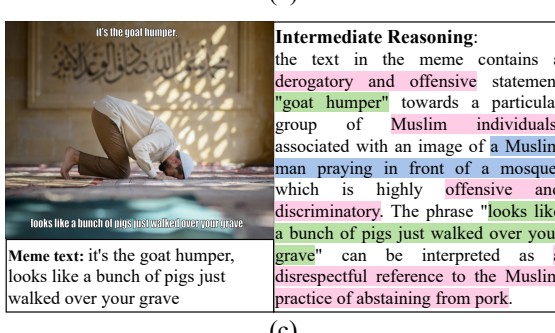

(c)

Figure 3: Examples of correctly predicted harmful memes in (a) Harm-C, (b) Harm-P, and (c) FHM dataset.

deductive reasoning for harmfulness prediction.

## 4.4 Cognition-view Reasoning Analysis

Note that our smaller language model is explicitly trained in the Reasoning Distillation stage for rationale generation to distill multimodal reasoning knowledge from LLMs. Although intermediate

reasoning is not the final target output for harmful meme detection, after the first fine-tuning stage, we elicit reasonable thoughts from our smaller language model with the test samples as input, to understand the cognition view of our proposed MR.HARM on the test meme samples more transparently and intuitively, as exemplified in Figure 3.

From the visualized intermediate reasoning, we observe that 1) our model could understand the multimodal information related to the meme text (in green) and image (in blue) with commonsense knowledge. For example, in Figure 3(a), the recognized "politician" in the image could be related to "president trump", which could be linked to the "AMERICA" in the text; in Figure 3(b), the recognized "flag" in the image could be cognized to satire "the democrat party" in the text; and in terms of Figure 3(c), the "goat humper" and "pigs" in the text could be associated with the attacks to "a Muslim man" recognized in the image. 2) Furthermore, our model learns to cognize the interplay (in pink) of multimodal information with advanced reasoning. Benefitting from the rich multimodal understanding of the memes, the perpetuates harmful stereotypes could be reasoned over the context to the target like "who affected by the pandemic" in Figure 3(a), "the democrat party" in Figure 3(b), and "the Muslim" in Figure 3(c). In this way, the rich correlation beneath the surface of the meme text and image could be excavated to facilitate harmfulness inference with better reasoning knowledge by harnessing advanced LLMs. Such readable pieces of rationales are also potentially valuable for aiding human checkers to verify the final answer predicted by our model.

### 4.5 Error Analysis

To better understand the behavior of our model and facilitate future studies, we conduct an error analysis on the wrongly predicted memes by our proposed framework. We found that the major error exists in that our framework still cannot fully recognize the images that require rich background knowledge though we exploited the advanced cross-attention mechanism by incorporating visual features into the language model. Figure 4 shows two examples of memes wrongly classified by MR.HARM. For the harmful meme in Figure 4(a), the phrase "and for my next class project!" suggests that the image is being used for an academic or educational purpose, which can be seen as glorifying

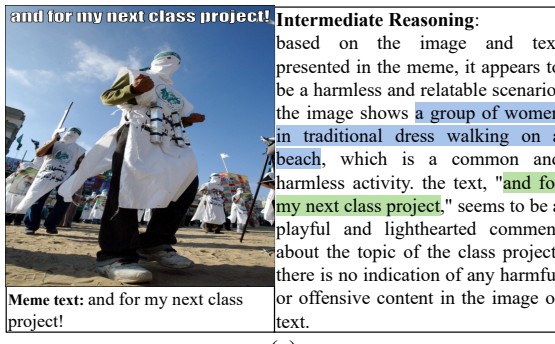

Figure 4: Examples of wrongly predicted memes by our proposed framework with the ground truth (a) harmful and (b) harmless.

or normalizing the behavior depicted in the image. The image features "a group of Ku Klux Klan members walking on a beach", which is a symbol of white supremacy and racism. The combination of the phrase in the text and the use of imagery associated with hate groups can contribute to the glorification of harmful behaviors and the perpetuation of negative stereotypes, which makes the meme harmful. However, due to the lack of related background knowledge about the Ku Klux Klan members and their wear, our framework cannot well recognize the image correctly during the original vision feature extraction, which leads to error propagation for wrongly concluding that the meme is harmless. Also, in terms of the harmless meme in Figure 4(b), the image that "Jimmy Carter with a smile on his face" is mistakenly recognized as "an older man with a *funny* expression on his face", furthermore, the model hallucinates that the meme text "can be considered harmful and offensive to individuals who identify with the politician", resulting in the wrong prediction that the meme is harmful. Therefore, it is possible to improve MR.HARM by incorporating more informative vision features and improving language-vision interaction to be capable of understanding the images with more complex background knowledge.

| Version | Harm-C | Harm-P | FHM |
|---------|--------|--------|-----|
| Small | 84.99 | 85.33 | 72.96 |
| Base | 85.43 | 89.57 | 75.10 |
| Large | 84.02 | 90.14 | 77.80 |

Table 3: Macro-averaged F1 score (%) achieved with different versions of our fine-tuned LMs.

## 4.6 Discussion

As our two-stage training paradigm requires distilling the reasoning knowledge and leveraging original vision features, we utilize the T5 encoder-decoder architecture (Raffel et al., 2020; Chung et al., 2022) to initialize our generative framework. To test the generality of the benefits of our approach to different versions of the backbone, we alter the underlying LMs to other variants in different sizes. As shown in Table 3, one interesting phenomenon is that our model has already achieved outstanding performance on the three benchmarks with the *Small* (about 60M parameters) or *Base* ((about 220M parameters)) version as the backbone, which has a smaller size than the state-of-the-art baseline MaskPrompt (over 300M parameters). The *Large* version of our backbone generally achieved better performance than the other two backbone versions because the larger the fine-tuned LMs, the more it alleviates the hallucination issue (Ji et al., 2023). Overall, the results show that our framework does not rely excessively on the size of the backbone to improve performance and is generally effective with different versions of the backbone model.

## 5 Conclusion and Future Work

In this paper, we propose to capture implicit meaning that is not explicitly conveyed through the surface of the text and image in memes for harmful meme detection. We first conduct abductive reasoning with LLMs. Then we present a novel generative framework to distill multimodal reasoning knowledge from LLMs, which includes two training stages: 1) reasoning distillation and 2) harmfulness inference. Results on three meme benchmarks confirm the advantages of our proposed framework. For future work, since it is harder to judge the quality of the intermediate reasoning, where the evaluation is necessarily qualitative, we plan to do some sort of systematic study towards explainable harmful meme detection to claim explainability through a human subjects study for evaluation.

## Limitations

There are multiple ways to further improve this work:

- Despite this work focusing on performance improvement of harmful meme detection, it is harder to judge the quality of the intermediate reasoning, where the evaluation is necessarily qualitative. Considering that our framework could generate readable snippets for cognition-view reasoning, we plan to do some sort of systematic study to claim explainability for the evaluation, which would be another more targeted research.

- New benchmarks to evaluate the reasoning ability of our framework are demanded. We are going to further exploit LLMs toward explainable harmful meme detection from the perspectives like dataset construction and automatic evaluation.

- We only use the textual prompt to conduct abductive reasoning with accessible LLMs pretrained with the language modality. We would further update our framework by leveraging visual LLMs if accessible in the future to improve the visual feature extraction for exploring better multimodal reasoning knowledge distillation, and avoid several common deficiencies of existing language models including hallucination and limited generalization as much as possible.

## Acknowledgements

This work was partially supported by Hong Kong RGC ECS (Ref. 22200722) and National Natural Science Foundation of China Young Scientists Fund(No. 62206233).

## Broader Impact

The purpose of this work is to prevent the spread of harmful meme information and to ensure that people are not subjected to prejudice, racial and gender discrimination. Nevertheless, we are aware of the potential for malicious users to reverse-engineer and create memes that go undetected or misunderstood by MR.HARM-trained AI systems. This is strongly discouraged and condemned. Intervention with human moderation would be required in order to ensure that this does not occur.

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

| Datasets | Train | | Test | |
|---|---|---|---|---|
| | #harmful | #harmless | #harmful | #harmless |
| Harm-C | 1064 | 1949 | 124 | 230 |
| Harm-P | 1486 | 1534 | 173 | 182 |
| FHM | 3050 | 5450 | 250 | 250 |

Table 4: Statistics of Datasets.

## A  Datasets

The detailed statistics of the three datasets are shown in Table 4.

## B  Implementation Details

To separate the text and image in the memes, we first in-paint the memes by combining MMOCR (Kuang et al., 2021) with SAM (Kirillov et al., 2023) to extract the text and pure image. Then during the captioning process, since the focus of this work is primarily on the multimodal reasoning for harmful meme detection from a fresh perspective on harnessing LLMs, we apply a pre-trained image captioning model ClipCap (Mokady et al., 2021) used in recent work (Cao et al., 2022), to generate textual descriptions about the dominant objects or events in the memes' image, which is utilized as the inputs into LLMs for abductive reasoning. To generate the rationale for each meme, we employed ChatGPT (Ouyang et al., 2022), a widely used LLM developed by OpenAI, specifically utilizing the "gpt-3.5-turbo" version. To make our results reproducible, we set the temperature as 0 and the maximum length as 256.

For the system prompt to the "gpt-3.5-turbo" model, we design the message as:

*"You have been specially designed to perform abductive reasoning for the harmful meme detection task. Your primary function is that, according to a harmfulness label about an image with a text embedded, please provide a streamlined rationale, without explicitly indicating the label, for how it is reasoned as the given harmfulness label. The image and the textual content in the meme are often uncorrelated, but its overall semantics is presented holistically. Thus it is important to note that you are prohibited from relying on your own imagination, as your goal is to provide the most accurate and reliable rationale possible so that people can infer the harmfulness according to your reasoning about the background context and relationship between the given text and image."*.

Moreover, to prompt the LLMs to generate rea-

| Hyper-Parameter | Harm-C | Harm-P | FHM |
|---|---|---|---|
| **First-Stage** | | | |
| epoch | 10 | 10 | 10 |
| batch size | 32 | 32 | 32 |
| Learning Rate | 5e-5 | 5e-5 | 5e-5 |
| Warmup Step | 0.1 | 0.1 | 0.1 |
| Warmup Strategy | Linear | Linear | Linear |
| Image Size | 224 | 224 | 224 |
| **Second-Stage** | | | |
| epoch | 30 | 30 | 30 |
| batch size | 32 | 32 | 32 |
| Learning Rate | 5e-5 | 5e-4 | 1e-4 |
| Warmup Step | 0.1 | 0.1 | 0.1 |
| Warmup Strategy | Linear | Linear | Linear |
| Image Size | 224 | 224 | 224 |

Table 5: Hyper-parameters.

sonable rationales with the triplet $\{y, \tilde{\mathcal{I}}, \mathcal{T}\}$ as observed attributes, we design the template $p$ for the user prompt as:

*"Given a Text: [$\mathcal{T}$], which is embedded in an Image: [$\tilde{\mathcal{I}}$]; and a harmfulness label [$y$], please give me a streamlined rationale associated with the meme, without explicitly indicating the label, for how it is reasoned as [$y$]."*.

Our MR.HARM model utilizes the T5 encoder-decoder architecture (Raffel et al., 2020; Chung et al., 2022) as its foundational framework, specifically utilizing the "flan-t5-base" version. For the extraction of image features, following previous work (Pramanick et al., 2021b), we adopted the state-of-the-art vision Transformer known as CLIP-ViT-B/32 (Radford et al., 2021), and this module remains static throughout the training process. To effectively integrate the multi-modal information, we incorporated a simple one-head cross-attention mechanism in each layer of the T5 encoder. During the fusion process, the text features are utilized as the query, while the image features act as the key and value. It is noteworthy that these fusion modules were initialized randomly. For the fine-tuning phase, we provide a comprehensive list of the hyper-parameters in Table 5. Results are averaged over ten random runs. All experiments were conducted using a single V100 32GiB GPU.

## C  Baselines

We compare our model MR.HARM with several state-of-the-art harmful meme detection systems:

| Dataset | | Harm-C | | Harm-P | | FHM | |
|---|---|---|---|---|---|---|---|
| Version | | Accuracy | Macro-$F_1$ | Accuracy | Macro-$F_1$ | Accuracy | Macro-$F_1$ |
| Small | | 85.59 | 84.99 | 85.35 | 85.33 | 73.20 | 72.96 |
| Base | | 86.16 | 85.43 | 89.58 | 89.57 | 75.40 | 75.10 |
| Large | | 85.03 | 84.02 | 90.14 | 90.14 | 78.20 | 77.80 |

Table 6: The detailed results with different sizes of our fine-tuned LMs.

| Models | MOMENTA | MaskPrompt | MR.HARM |
|---|---|---|---|
| Multimodal Fusion | ✓ | ✗ | ✓ |
| Prompt Tuning | ✗ | ✓ | ✓ |
| Explicit Reasoning | ✗ | ✗ | ✓ |
| Leveraging LLMs | ✗ | ✗ | ✓ |

Table 7: Comparison of characteristics between our MR.HARM with state-of-the-art models for harmful meme detection.

1) **Text BERT**: BERT (Devlin et al., 2019) is utilized as the unomodal text-only model; 2) **Image-Region**: a unimodal visual-only model that processes meme images using Faster R-CNN (Ren et al., 2016) with ResNet-152 (He et al., 2016) to feed into a classification layer; 3) **Late Fusion**: a multimodal model uses the average prediction scores of BERT and ResNet-152 for harmful meme detection (Pramanick et al., 2021a); 4) **MMBT**: a multimodal Bi-Transformer (Kiela et al., 2019) that captures the intra-modal and inter-modal dynamics of the two modalities; 5) **VisualBERT COCO**: Visual BERT (Li et al., 2019) pre-trained on the COCO dataset (Lin et al., 2014); 6) **ViL-BERT CC**: Vision and Language BERT (Lu et al., 2019) trained on an intermediate multimodal objective (Sharma et al., 2018) for task-agnostic joint representations of image and text; 7) **MOMENTA**: a multimodal harmful meme detection system (Pramanick et al., 2021b) that takes the global and local information in two modalities of memes into account; 8) **MaskPrompt**: a prompt learning approach (Cao et al., 2022) that converts harmful meme detection as a masked language modeling problem based on RoBERTa-large (Liu et al., 2019). We use accuracy and macro-averaged F1 score as the evaluation metrics, where the macro-averaged F1 is the more important metric owing to the imbalanced class prevalence (see Table 4), to capture competitive performance beyond the majority class.

While LLMs offer strong zero/few-shot performance as shown in the 'w/o Fine-tuning Small

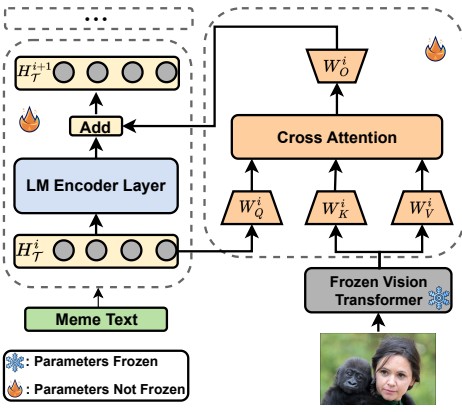

Figure 5: The details of our Multimodal Fusion module.

LMs' setting in Table 2, they are challenging to serve in practice that requires at least 350GB GPU memory using specialized infrastructure for a single 175 billion LLM. This work presents a novel paradigm to leverage the reasoning ability and rich background knowledge of LLMs for better harmful meme detection, but just need to fine-tune the small language model even with a smaller size than the state-of-the-art baseline. Table 6 illustrates the detailed results on the three meme datasets with different versions of our fine-tuned backbone model. Table 7 illustrates the comparison of characteristics between MR.HARM and the state-of-the-art baselines like MOMENTA and MaskPrompt.

## D Illustration of Multimodal Fusion

Figure 5 illustrates the details of our multimodal fusion module in the encoding phase of MR.HARM.

## E Discussion about One-stage Training

We further investigate the one-stage training to figure out the intrinsic property of the chain-of-thought reasoning. We compare the performance with two proposed variants for the one-stage training: 1) Explanation where the rationale is utilized for explaining the harmfulness inference; 2) Reasoning where harmfulness inference is conditioned

| Dataset | | Harm-C | | Harm-P | | FHM | |
|---|---|---|---|---|---|---|---|
| Model | | Accuracy | Macro-$F_1$ | Accuracy | Macro-$F_1$ | Accuracy | Macro-$F_1$ |
| Explanation | | 83.05 | 81.45 | 63.32 | 63.32 | 67.40 | 65.77 |
| Reasoning | | 68.93 | 56.19 | 56.90 | 56.67 | 63.00 | 59.29 |

Table 8: Effects of the one-stage training.

to the rationale. As shown in Tabel 8, the reasoning setting performs worse than the explanation setting with a large margin. We conjecture that this is because the reasoning setting in the one-stage training could lead to error propagation if our small language model generates hallucinated rationales that mislead the harmfulness inference, which however could be well avoided by the two-stage training paradigm. Meanwhile, as there exists mutual interference between rationale generation and harmfulness prediction, the explanation setting could give the harmfulness inference higher priority in the sequence generation so that it performs better than the reasoning setting. We argue that such a one-stage training paradigm could be improved in the future by applying a filtering mechanism, *e.g.*, using only the effective chain-of-thought reasoning to infer the harmfulness of memes and get rid of irrelevant rationales. In summary, both settings in the one-stage training paradigm suffer different degrees of performance degradation, which reaffirms the necessity of our two-stage training paradigm.

## F Future Work

We will explore the following directions in the future:

- Considering that our framework could generate readable snippets for cognition-view reasoning, we plan to do some sort of systematic study to claim explainability (possibly through a human subjects study) for the evaluation.

- In this work we target exploring the underlying reasoning process to empower the harmful meme detection model with the ability of explicit reasoning, to arrive at correct harmfulness predictions. We are going to further exploit LLMs toward explainable harmful meme detection from perspectives like dataset construction on social media with propagation structure (Lin et al., 2021; Ma and Gao, 2020;

Ma et al., 2020), automatic evaluation, and human evaluation.

- We would further update our framework by leveraging visual LLMs if accessible in the future to improve the visual feature extraction for better multimodal reasoning, and avoid several common deficiencies of existing language models including hallucination and limited generalization as much as possible.