# OpenReview forum: "Beneath the Surface: Unveiling Harmful Memes with Multimodal Reasoning Distilled from Large Language Models"
_EMNLP/2023/Conference — EMNLP 2023 Findings_

### Official Review · Reviewer_3phz · 2023-08-02

**Typos Grammar Style And Presentation Improvements:** The presentation is fine.
**Soundness:** 4

**Ethical Concerns:**

Yes

**Excitement:**

3: Ambivalent: It has merits (e.g., it reports state-of-the-art results, the idea is nice), but there are key weaknesses (e.g., it describes incremental work), and it can significantly benefit from another round of revision. However, I won't object to accepting it if my co-reviewers champion it.

**Justification For Ethical Concerns:**

Due to the focus of this work on detecting harmful memes in social media, there may be some discriminatory or harmful content.

**Missing References:**

None

**Paper Topic And Main Contributions:**

This paper investigates the problem of detecting harmful memes, which is of great significance in today's increasingly popular social media platforms. Compared to previous end-to-end approaches for detecting harmful memes, this work proposes a two-stage training framework based on a large-scale language (LLM) model. The framework aims to improve multimodal fusion and lightweight fine-tuning for predicting harmfulness. It involves distilling multimodal reasoning knowledge from the LLM and fine-tuning a generative model for harmfulness inference. The authors highlight that this work is the first to explicitly leverage the knowledge of LLM to address the issue of superficial understanding of harmful memes. The effectiveness of the proposed model is validated through experiments on three real-world datasets.

**Questions For The Authors:**

See the previous section.

**Reasons To Accept:**

1. The research topic is important.

2. The method of integrating LLM is straightforward but effective. The experiments are comprehensive.

3. The paper is well-structed and easy to follow. Moreover, the authors provide many case studies for better understanding.

**Reasons To Reject:**

1. All the techniques used in the study are already existing and well-known, and the author has cleverly integrated them. In my opinion, the biggest contribution of this work seems to be how it introduces LLM into this focused task.

2. The baselines compared by the author are not based on LLM, and I am not sure if this comparison is fair. It remains unknown how these representative baseline models would perform if equipped with LLM.

3. The author does not provide reproducible source code.

**Reproducibility:**

3: Could reproduce the results with some difficulty. The settings of parameters are underspecified or subjectively determined; the training/evaluation data are not widely available.

**Reviewer Confidence:**

3: Pretty sure, but there's a chance I missed something. Although I have a good feel for this area in general, I did not carefully check the paper's details, e.g., the math, experimental design, or novelty.

---

> ### Author Rebuttal · Authors · 2023-08-27
>
> Thank you for valuable comments. We are encouraged that the reviewer found our research to be important, with the effective method, comprehensive evaluations and well-structured writing. In response to the reviewer's feedback, we have diligently addressed all of the concerns. We are committed to making the necessary modifications and additions to improve our paper in the subsequent version.
>
> **Q1**: All the techniques used in the study are already existing and well-known, and the author has cleverly integrated them. In my opinion, the biggest contribution of this work seems to be how it introduces LLM into this focused task.
> **A1**: Thank you for acknowledging that, *exploring harmful meme detection from a fresh perspective on advanced LLMs* is a main contribution of this work. However, this is **not** the only contribution of our paper. As depicted in Sec.1, our contributions are underestimated in the following aspects:
> 1) We identify the key limitation of existing harmful meme detection methods for understanding memes (Line#065-102);
> 2) We propose to conduct abductive reasoning to activate reasoning knowledge related to gold-truth labels from LLMs (Line#266-282);
> 3) We propose the knowledge distillation strategy to inject reasoning knowledge from single-modality to multimodal inputs (Line#319-347);
> 4) We propose the design philosophy of the two-stage training paradigm on small LMs, complementary to each other with the abductive reasoning step (Sec.3.2-3.3);
> 5) We conduct extensive experiments with three public meme benchmarks, and demonstrate that our proposed method achieves much better performance than the newest SoTA harmful meme detection baselines.
>
> We believe that all these contributions amount to a major departure compared to SoTA, both algorithmically and practically speaking.
>
> **Q2**: How these representative baseline models would perform if equipped with LLM?
> **A2**:
> 1) First of all, to equip existing SoTA harmful meme detection methods with the reasoning knowledge from LLMs, one straightforward and possible solution would be: use the rationale generated from LLMs as input for an end-to-end classification training. By this design, there may be two inherent weaknesses:
> (I) With just the meme text and image caption as prompt of LLMs, the generated rationales are always noisy and low-quality, even towards the wrong label. It could directly decrease the final detection performance (Line#515-519);
> (II) If without our proposed abductive reasoning mechanism, the quality of generated rationales from LLMs needs some additional designs to filter; If with the abductive reasoning mechanism, baselines cannot be deployed for model testing as stated in Line#300-302.
> 2) For more comprehensive comparisons, we also conducted the ablative studies about our models without LLMs. As shown in Table 2 (‘w/o Reasoning Distillation’), it proves that our model can still achieve competitive performance compared with baselines, even discarding that strategy with LLMs. Please refer to Table 1 and Table 2 for more details. In Figure 4, we further analyzed that our framework has fewer training parameters than some of the SoTA baselines.
>
> **Q3**: The author does not provide reproducible source code.
> **A3**: Our approach could be easily reproduced according to Sec.3 (Line#266-389) and Implementation Details (Line#911-975). We will release all the source codes once published (Line#150).

---

### Official Review · Reviewer_f6a5 · 2023-08-04

**Soundness:** 3

**Excitement:**

2: Mediocre: This paper makes marginal contributions (vs non-contemporaneous work), so I would rather not see it in the conference.

**Missing References:**

-Line 357: generated by LLMs (Sutskever et al., 2014)
Is this reference appropriate? Please recheck.


**Paper Topic And Main Contributions:**

The paper presents a 2-stage framework for harmful meme detection, with reasoning distillation as part of the first step and harmfulness inference as the second. The first stage involves fine-tuning an encoder-decoder setup for generating intermediate reasoning, which is obtained via the abductive reasoning from the LLMs. The setup includes a multimodal fusion strategy for getting encoder representations, which are then used for conditioning towards generating intermediate reasoning text. The second stage uses the same modelling setup, with a different decoder object configured towards generating the harmfulness type. The authors have compared the performance of the proposed approach on three datasets (Harm-P, Harm-C and FBHM) and showcase its superiority over other competitive baselines.

The paper contributes via the following:
- This is one of the initial works exploring the utility of text-based LLMs towards a 2-stage multimodal task.
- A novel lightweight generative setup addressing the complex task of harmfulness prediction for memes.
- Presents ablation studies to corroborate the performance edge of the proposed approach.


**Questions For The Authors:**

[A]. Lines 286-289: “...and  invalid rationals are naturally filtered out.”
Were the hallucinations completely taken care of?

[B]. Eq 2: Are you proposing a “cross-attention” based approach or a “co-attention” one? You’ve shown equations for computing attended visual representations conditioned upon the textual ones. Do you do it similarly the other way around as well? Maybe it's not that obviously clear from the details provided; please provide details and clarify.

[C]. Appendix F (Error Analysis, Fig. 6)
-  Example (a): Did you try to analyze the implications of various visual artefacts like image quality, occlusion, obscurity, object category, etc., while correlating them with the likely errors that the proposed model made? If yes, adding details on that assessment would be more insightful (in the Appendix, maybe).
-  Example (b): Did you consider the possibility of likely wrong annotations within the dataset and maybe analyzing the stances adopted by the LLM while generating responses on a case-to-case basis (at least for a few samples)? As for all we know, the political reputation and electoral prospects might get severely affected due to such memes, which could potentially contribute to harmfulness.
-  Lines 1090-1093: the image “Jimmy Carter with a smile on his face” is mistakenly recognized as “an older man with a funny expression on his face.”
How do we ascertain that it's a mistake on LLM’s part? In fact, such an interpretation of Jimmy Carter’s expression could be leveraged to infer the potentially intended harmfulness within the meme.


**Reasons To Accept:**

The task is complex and relevant, and the findings would strive to enhance our current understanding of the capacity/utility of LLMs for multimodal tasks, as against the conventional paradigm of exploring early/late, single/dual-stream multimodal modeling approaches.

**Reasons To Reject:**

1. The write-up quality needs to improve significantly.
2. Given the nature of the modularised approach proposed, the paper lacks sufficient evaluations w.r.t the datasets/tasks explored. Additional task categories from amongst emotions, offensiveness and sexism, for which publicly accessible meme datasets are available, should have been studied to establish better generalizability of the proposed approach.
3. The analysis section could be further strengthened.


**Reproducibility:**

4: Could mostly reproduce the results, but there may be some variation because of sample variance or minor variations in their interpretation of the protocol or method.

**Reviewer Confidence:**

5: Positive that my evaluation is correct. I read the paper very carefully and I am very familiar with related work.

**Typos Grammar Style And Presentation Improvements:**

-Lines 165-167: “...and struggle to yield good performance only using unimodal detection methods”.
Sure, but what does? Is it the researchers in general, models, LLMs, SOTA or conventional techniques? Proper Co-referencing seems to be missing.

-Lines 286-289: “Because the rich contextual background knowledge could be activated by  abductive reasoning based on the ground truth and  invalid rationals are naturally filtered out.”
Could be rephrased?

-Line 409, 977: sevral -> several

-Line 646-648: Disclaimer: “This paper contains discriminatory content that may be disturbing to some readers.”
It would be prudent to pull this to the first page rather than at the end for the required pretext.

-Line 314: “...generated rationales from LLMs”
It's possible that I might’ve missed it, but is the specific LLM (ChatGPT) used for generating rationales mentioned anywhere before line 484 (Ablative study, Page 6)? I see the details regarding the implementation, base model used etc. are elaborated in the Appendix, but at least the basic details regarding these primary components/technologies leveraged/considered must be mentioned (or atleast referred to) as part of the main paper wherever relevant, so as enable better understanding of the scope and implications of the experimental choices/decisions made, w.r.t the modeling setup, and hence the results.

---

> ### Author Rebuttal · Authors · 2023-08-27
>
> Thank you for your meticulous feedback. We are delighted by the reviewer's recognition of the soundness, promotive findings, and distinctiveness of our research.  We've responded to all the concerns and will accordingly modify and supplement our paper in the subsequent version.
>
> ***Major Concerns***:
> **Q1**: The paper lacks sufficient evaluations w.r.t the datasets/tasks explored from amongst emotions, offensiveness and sexism.
> **A1**:
> 1) We have conducted the evaluations on the hateful emotion in memes (**FHM** dataset) as one of the additional task categories in harmful meme detection. Specifically, we empirically evaluated our work on three public benchmarks: the first two datasets (Harm-C, Harm-P) are benchmarks for harmful meme detection defined by previous work [1, 3]; the later one (FHM) is a meme dataset [2, 4] related to emotions with hate speech (Line#399), which is one of the additional task categories that the reviewer#f6a5 concerned, achieving 9.86\% improvement of Mac-F1 score over the SoTA baseline[2].
> 2) As emphasized in Sec.4.1, we strictly followed the dataset and evaluation settings in previous work [1, 2, 3] on harmful meme detection, where ***harm** [3] is defined much broader and generalized than hate emotions [4], offensiveness [5], and sexism [6]*.
> 3) To further address this concern of Reviewer#f6a5, we further evaluate our model on MultiOff [5] and MAMI [6] datasets from offensiveness and sexism perspectives. Our model could achieve Mac-F1 score (improvement over [2]) of 69.61\% (3.51\%) and 74.98\% (4.03\%) on MultiOff and MAMI datasets.
>
> [1] Pramanick, Shraman, et al. "MOMENTA: A Multimodal Framework for Detecting Harmful Memes and Their Targets." Findings of the Association for Computational Linguistics: EMNLP 2021. 2021.
> [2] Cao, Rui, et al. "Prompting for Multimodal Hateful Meme Classification." Proceedings of the 2022 Conference on Empirical Methods in Natural Language Processing. 2022.
> [3] Pramanick, Shraman, et al. "Detecting Harmful Memes and Their Targets." Findings of the Association for Computational Linguistics: ACL-IJCNLP 2021. 2021.
> [4] Kiela, Douwe, et al. "The hateful memes challenge: Detecting hate speech in multimodal memes." Advances in neural information processing systems 33 (2020): 2611-2624.
> [5] Suryawanshi, Shardul, et al. "Multimodal meme dataset (MultiOFF) for identifying offensive content in image and text." Proceedings of the second workshop on trolling, aggression and cyberbullying. 2020.
> [6] Fersini, Elisabetta, et al. "SemEval-2022 Task 5: Multimedia automatic misogyny identification." Proceedings of the 16th International Workshop on Semantic Evaluation (SemEval-2022). 2022.
>
>
> **Q2**: The analysis section could be further strengthened.
> **A2**: Thank you for your feedback. We have made the comprehensive analysis in our submission, including: 1) Detection performance analysis (Sec.4.2); 2) Ablative analysis (Sec.4.3) about our contributions; 3) Case studies (Sec.4.4) for the intermediate reasoning of small LMs; 4) Analysis of model size and training parameters (Sec.4.5) for small LMs; 5) Training paradigm analysis (Sec.E); 6) Error analysis (Sec.F) and 7) Future Work (Sec.I), etc.
> But unfortunately, due to limited space, we put more analysis and data examples in Appendix. We will mark the correspondence between Appendix and the main paper more explicitly, and try to discuss more error analysis in revision.
>
> ***Minor Concerns***:
> **Q3**: The write-up quality and typo improvement issue.
> **A3**: We would further polish our writing in the subsequent version.
>
> **Q4**: Line 357: generated by LLMs (Sutskever et al., 2014) Is this reference appropriate? Please recheck.
> **A4**: Yes. The reference is for the whole sentence (Line#355-357) depicting the loss in the sequence-to-sequence learning between the predicted text and the rationale generated by LLMs, not only for Line#357. We will rewrite this sentence to mitigate misunderstanding.
>
>
> ***Open Questions***:
> **Q5**: Were the hallucinations completely taken care of? (Question A)
> **A5**: The original sentence in the manuscript is “*the hallucination issue of LLMs could be **effectively alleviated**. Because the rich contextual background knowledge could be activated by abductive reasoning based on the ground truth and invalid rationales are naturally filtered out.*”
> 1) Except for the reasoning knowledge based on the ground truth, the invalid rationales for the other labels (not ground truth) are completely filtered out with the abductive reasoning mechanism.
> 2) Besides the abductive reasoning mechanism, our further proposed two-stage training paradigm enables that, only the final fine-tuned model after the Harmful Inference stage is applied for the model testing (Line#387-389), so there is no explicit hallucination issue during testing because the model only generates the predicted label for harmful meme detection.
>
> **Q6**: You’ve shown equations for computing attended visual representations conditioned upon the textual ones. Do you do it similarly the other way around as well for the cross-attention? (Question B)
> **A6**: As stated in Eq.(3), it can only do it as Eq.(2), and obviously **not work** if do it similarly the other way around:
> 1) From the motivation perspective: We have clarified in the justification (Line#311-318), our goal is to fine-tune the small language model.  So we need to train the language model encoder while the vision Transformer is frozen in the Encoding part (Line#323-332). To this end, Eq.(2) is proposed to attend the visual representations to the textual ones as the input of each LME layer.
> 2) From the technique perspective: If we do computing attended textual representations conditioned upon the visual ones, the output dimension would be $\bf n \times d$ (Line#332), which is incompatible with the dimension $\bf m \times d$ of each LME layer (Line#324) and then causes Eq.(3) wrong during computing.  Besides, we also presented a more intuitive illustration (Figure 5) of the cross-attention mechanism in Appendix Sec.D.
>
> **Q7**: Discussion about Appendix Sec.F (Figure 6).
> **A7**:
> 1) Example (a):
> We exemplify two more wrongly predicted examples for error analysis of the implications of some visual artifacts:
> (**Examlpe I**) A test example covid_memes_5450 in Harm-C dataset, which is similar to Figure 9.b in Appendix. The meme text is “*Me\nTrump with\ncovid19\nIs this karma?*”. After the first fine-tuning stage (Reasoning Distillation), we generated the intermediate reasoning:
> “*based on the text and image caption, it appears that the meme is making a political statement about trump's covid-19 covid-19 treatment. the text suggests that trump is a political figure, which could be interpreted as a reference to the current covid-19 pandemic. the image seems to be unrelated to the text and could be interpreted as a reference to the current covid-19 pandemic. ...*”
> In the harmful meme, the text “Me” refers to the man wearing glasses, standing in front of a window; the text “Trump with\ncovid19” refers to the butterfly; the man comments that “Is this karma?” on the butterfly. However, the object category like butterfly is not correctly recognized due to the low-quality image (obscurity), and the inter-relationship between the image with the meme text is still not cognized, which leads to the generated reasoning making no sense.
> (**Example II**) One more harmful test example is covid_memes_5644,  the girl dressed in red is in obscurity on her face, and the image is also low-quality. Character relationships in the image are not captured due to the occlusion of the American flag.
> So the generated intermediate reasoning is that “*...the image shows a man in a suit and tie, which is a popular and popular show. the text suggests that the person is going golfing and ignoring the pandemic, ... the meme does not appear to contain any offensive or harmful content and is likely intended to be a lighthearted joke*". Therefore, in future work (Line#1133-1139), we proposed to leverage more powerful vision LLMs in the reasoning distillation stage of model training to improve the robustness of our small LMs in model testing.
> 2) Example (b):
> - Possible wrong annotations?
>   · Yes. We also found there are some wrong annotations. Specifically, in rare cases with wrong annotations, although we conduct abductive reasoning with the wrong golden label (Note that the LLMs can only be used during model training as clarified in Line#299-304), the content generated by LLMs still cannot well indicate how the meme is reasoned as the gold truth, like just output “the meme seems to be promoting a harmful and potentially harmful message” but without any concrete reasonable thought chains about the conclusion. On a positive note, the phenomenon illustrates that the lack of logical reasoning knowledge in LLMs towards wrong annotations, which also contributes to exploring better robustness of our approach to some extent for the wrongly labeled data.
> - How do we ascertain that it's a mistake on LLM’s part?
>   · Note that there is no any statement indicating it’s a mistake on LLM’s part in Sec.F. It’s a mistake generated by our small language model during testing. It is a test sample while LLMs can only be used during model training as clarified in Line#299-304.
>   · In the image, Jimmy Carter is not making a funny face. And it is not a meme text. The example about “funny expression” could be referred to the memes_44 or memes_8240 (Figure 10.b) in Harm-P dataset. And the “Jimmy Carter with a smile on his face” is the caption we used the captioning models like ClipCap and BLIP on the test sample to generate for analysis of the error result. Besides, as human beings, we cannot get any funny elements on his face, so we do not easily identify it as a wrong annotation.
>   ·  Thus it is important to note that the models are prohibited from relying on their own imagination. Any interpretation or reasoning needs to be built on the correct recognition of the harm-indicative patterns.

---

### Official Review · Reviewer_ue2E · 2023-08-04

**Soundness:** 4

**Excitement:**

3: Ambivalent: It has merits (e.g., it reports state-of-the-art results, the idea is nice), but there are key weaknesses (e.g., it describes incremental work), and it can significantly benefit from another round of revision. However, I won't object to accepting it if my co-reviewers champion it.

**Paper Topic And Main Contributions:**

This work presents a significant contribution to the field of harmful meme detection in social networks. The author proposes a novel multimodal framework, leveraging the reasoning capabilities of LLMs, to effectively identify harmful memes with robust support. The model's training methodology involves a two-stage strategy, starting with the distillation of multimodal reasoning from LLMs, followed by fine-tuning the generative framework to infer harmfulness.
The main contributions of the model are:
1. This work introduces a pioneering approach by incorporating the reasoning power of LLMs, such as ChatGPT, into the task of harmful meme detection. This innovative integration of advanced language models enables the model to effectively reason about and identify harmful content within memes, enhancing the overall detection accuracy.
2. By leveraging multimodal information, intrinsic to memes, in the reasoning process, the model successfully captures and encodes the complex and integrated semantic information present in the memes. This multimodal reasoning approach proves beneficial in effectively understanding the contextual nuances and visual cues that contribute to the harmfulness of memes.
3. The proposed two-stage training strategy, involving the distillation of reasoning knowledge from LLMs followed by fine-tuning on specific tasks, is a notable contribution to the field. This approach optimizes the model's performance by focusing on the harmful meme detection task while also mitigating the computational costs associated with using LLMs, both in terms of time and space.
Overall, this paper presents a well-developed and well-structured framework for harmful meme detection, incorporating innovative elements of LLM reasoning and multimodal information processing. The experiments and evaluations conducted by the author demonstrate the effectiveness of the proposed model in accurately identifying harmful memes within social networks. The insights provided by this work are of great value to the research community in the domain of Natural Language Processing and content moderation in social media platforms.

**Questions For The Authors:**

Question A:
Why distilling can help alleviate the hallucination issue?
Although the authors conduct the ablation test without distilling to demonstrate its necessity, the authors should cite related research or provide more in-depth analysis or experiments to demonstrate how the reduction of parameters and limiting knowledge through distillation can effectively alleviate hallucination issues.

**Reasons To Accept:**

1. The language used in the manuscript is clear and comprehensible, effectively conveying the ideas and concepts presented.
2. The chosen task of harmful meme detection is of utmost importance for social networks, given the pervasive influence of memes in online conversations. The authors have adeptly harnessed the reasoning capabilities of LLMs to enhance model interpretability and achieve superior performance compared to other SOTA methods.
3. The proposed model is presented in a lucid manner, making it easily understandable and replicable for researchers in the field. Furthermore, the framework's versatility allows for its straightforward adaptation to various other NLP tasks.
4. The conducted experiments are extensive and robust, effectively showcasing the efficacy of the proposed model. The inclusion of ablation tests highlights the significance of all individual components, and the detailed analysis provides compelling examples of multimodal reasoning. Moreover, the model's performance with flexible parameters is competitive with other existing models.

**Reasons To Reject:**

1. The proposed model relies on distilling multimodal reasoning from ChatGPT without applying any filtering mechanism. This may lead to a significant impact on the model's performance, as the quality of reasoning from ChatGPT may vary. It is crucial for the authors to address this issue and explore ways to ensure that only high-quality reasoning is utilized in the proposed model.
2. There might be a gap between the usage of images in the prompts and the image features used in the model. The authors should clarify how the multimodal information from images is integrated into the reasoning process and discuss any possible misalignments or discrepancies that might arise due to this disparity.
3. Although the authors have conducted tests on benchmark datasets of memes, the datasets themselves could have limitations and might become outdated over time. Given the dynamic and rapidly changing nature of memes on the internet, there is a concern regarding whether the model's effectiveness can be maintained in real-world scenarios. The authors should acknowledge this potential limitation and discuss strategies to keep the model up-to-date with evolving meme trends.
4. The selection of baselines may be limited, as the newest SOTA model is "MaskPrompt" from arxiv. To enhance the comprehensiveness and robustness of the evaluation, it is suggested that the authors incorporate a broader range of models published in the last two years. This inclusion should encompass both detection models and multimodal models based on LLMs. By incorporating a diverse set of baselines, the study can provide a more comprehensive evaluation and draw more insightful comparisons between the proposed method and SOTA alternatives.

**Reproducibility:**

4: Could mostly reproduce the results, but there may be some variation because of sample variance or minor variations in their interpretation of the protocol or method.

**Reviewer Confidence:**

4: Quite sure. I tried to check the important points carefully. It's unlikely, though conceivable, that I missed something that should affect my ratings.

---

> ### Author Rebuttal · Authors · 2023-08-27
>
> Thank you for insightful and careful comments. We are grateful for the reviewer's positive assessment of our research as novel work, with clear writing, and effective results. We have taken the valuable and constructive feedback into consideration and have tried to address your concerns. We will make the necessary modifications and additions accordingly.
>
>
> **Q1**: Explore ways to ensure that only high-quality reasoning is utilized in the proposed model?
> **A1**: This was indeed a tricky issue for us to consider in this work, for which we have proposed both the abductive reasoning mechanism and the two-stage training paradigm to ensure the quality of reasoning utilized in our proposed model.
> 1) Abductive reasoning with LLMs: We applied abductive reasoning on LLMs as the filtering mechanism in Sec.3.1, which constrains the direction of the multimodal reasoning just towards the gold-truth label in training data. This design can *explicitly ensure the generation of more reasonable rationale from LLMs*. For example, if one meme in training data is labeled as harmful, the invalid rationales for the wrong prediction (harmless) would be filtered out in this abductive reasoning manner.
>
> The results and analysis (Line#515) in Sec.4.3 further verified that if we only used LLMs without any filtering mechanism like abductive reasoning, the detection performance would largely decrease, which reaffirmed the effectiveness of abductive reasoning with the gold-truth label.
>
> 2) Two-stage training paradigm for small LMs: Meanwhile, we also proposed a two-stage training paradigm, which *not only addresses the issue that abductive reasoning is not applicable to model testing, but also further alleviates the negative impact imposed by low-quality reasoning from LLMs*. Even if for some challenging cases, the quality of reasoning from ChatGPT is “relatively” poor for distillation in the first stage, the second stage implicitly ensures that only high-quality reasoning knowledge can be induced and activated during backward fine-tuning (Line#378).
>
> In the analysis (Line#479; Line#1039) in (Sec.4.3; Sec.E), we also compared our two-stage training paradigm with a naïve one-stage one, which pointed out that the two-stage design is more robust than the one-stage one. In our two-stage training paradigm, the backward fine-tuning in the Harmfulness Inference stage would make it activate the harm-indicative knowledge absorbed in the Reasoning Distillation stage, unlike in the one-stage fashion letting the output of LLMs directly affect the prediction results.
>
>
> **Q2**: Why distilling can help alleviate the hallucination issue? (Question A)
> **A2**: Thank you for your question. First of all, we need to clarify that distilling (Sec.3.2) is not to alleviate the hallucination issue. It is a way to inject the multimodal reasoning knowledge from LLMs into our small model. More specifically, as we applied the abductive reasoning mechanism with LLMs to generate more reasonable rationales, the distilling process could equip our small model with the capability of multimodal reasoning without the label (Line#255-265, Line#299-301). After the distilling process, we have a second fine-tuning stage, Harmfulness Inference, to further circumvent the effect of the still-existing hallucination issue on the model's final predictions. As stated in Line#387-389, for the model testing, we only applied the final fine-tuned model after the Harmfulness Inference stage.
>
> **Q3**: The newest SOTA model is "MaskPrompt" from arxiv. Broader LLM-based models need to be selected as baselines.
> **A3**:
> 1) To the best of our knowledge, the latest SoTA model for harmful meme detection was MaskPrompt before this submission. And the MaskPrompt is accepted by EMNLP’22. We will revise the reference format in revision.
> 2) For the LLM-based model, we evaluated ChatGPT (‘gpt-3.5-turbo’)  (Line#483) and different training paradigms (like the reasoning setting in Line#1033 and explanation setting in Line#1035) on this task in the analysis (Sec.4.3; Sec.E) for a broader comparison. As for other contemporaneous works based on LLMs, which were all released after April, and we didn’t compare with them in our initial submission. In the subsequent version, we will try to contain more latest methods for a more complete comparison.
> 3) To further address this concern of Reviewer#ue2E, we directly applied the novel vision LLM, LLaVA (released in April 2023), as one multimodal baseline based on LLMs. The Accuracy and Mac-F1 scores are (50.28\%, 49.40\%), (49.84\%, 34.86\%), and (51.20\%, 46.51\%) on Harm-C, Harm-P and FHM, respectively. While our method could achieve (86.16\%, 85.43\%), (89.58\%, 89.57\%), (75.40\%, 75.10\%) accordingly. We can see from the evaluations of broader LLM-based models [ChatGPT] and [LLaVA] that, the direct deployment of both detection models and multimodal models based on LLMs, struggles without lightweight design specific to this task, which is claimed as one main contribution in our paper (Line#141-146).
>
> **Q4**: How the multimodal information from images is integrated into the reasoning process, and any possible misalignments or discrepancies that might arise due to this disparity?
> **A4**:
> 1) Mitigate information loss: We used the image captions in the prompts because LLMs are pre-trained in language modality (Line#294-299). To mitigate the information loss of the image captioning, we used original image features instead of captions in the two-stage training paradigm.
> 2) Avoid error propagation: Furthermore, in the second fine-tuning stage for harmfulness inference, we only relied upon the original visual features for multimodal fusion with the textual features. The possible misalignments, like the information loss in image captioning, would not directly affect the second fine-tuning stage. As stated in Line#387-389, for the testing, we only applied the final fine-tuned model after the Harmfulness Inference stage.
> 3) Experimental Analysis: The ablative results (Table 2) for ‘w/o multimodal fusion’ further verified the effectiveness of original image features in mitigating information loss. The experimental results in Table 2 and Table 8 have analyzed the effectiveness of our two-stage paradigm in avoiding error propagation.
>
> **Q5**: The datasets themselves might become outdated over time?
> **A5**: It’s a good question. 1) We acknowledge that the distribution drift in datasets over time is a potential limitation for almost all data-driven tasks. 2) However, the contribution of this work is proposing a novel paradigm to distill commonsense reasoning knowledge for the harmful meme detection task. The proposed framework is general enough, which should still work with newly released stronger LLMs or new data/memes appearing on the Internet. 3) As you suggested, we would further discuss that issue in the Limitation section in the revision.

---

### Meta-Review · Area_Chair_RJGo · 2023-09-19

**Recommendation:** 3

**Metareview:**

The paper introduces a 2-stage framework for harmful meme detection, leveraging the reasoning capabilities of Large Language Models (LLMs). Reviewers recognized this as an early exploration into using text-based LLMs for a multimodal task, praising the proposed two-stage training strategy involving knowledge distillation from LLMs and fine-tuning for specific tasks. However, reviewers expressed some concerns, which the authors addressed during the rebuttal stage and committed to resolving in the final version.

---

### Decision · Program_Chairs · 2023-10-07

**Decision:**

Accept-Findings

**Comment:**

The paper introduces a 2-stage framework for harmful meme detection, leveraging the reasoning capabilities of Large Language Models (LLMs). Reviewers recognized this as an early exploration into using text-based LLMs for a multimodal task, praising the proposed two-stage training strategy involving knowledge distillation from LLMs and fine-tuning for specific tasks. However, reviewers expressed some concerns, which the authors addressed during the rebuttal stage and committed to resolving in the final version.